# The Grasping Test Revisited: A Systematic Review of Functional Recovery in Rat Models of Median Nerve Injury

**DOI:** 10.3390/biomedicines10081878

**Published:** 2022-08-03

**Authors:** Henrik Lauer, Cosima Prahm, Johannes Tobias Thiel, Jonas Kolbenschlag, Adrien Daigeler, David Hercher, Johannes C. Heinzel

**Affiliations:** 1Department of Hand-, Plastic, Reconstructive and Burn Surgery, BG Unfallklinik Tuebingen, University of Tuebingen, Schnarrenbergstraße 95, 72076 Tuebingen, Germany; hlauer@bgu-tuebingen.de (H.L.); cprahm@bgu-tuebingen.de (C.P.); jthiel@bgu-tuebingen.de (J.T.T.); jkolbenschlag@bgu-tuebingen.de (J.K.); adaigeler@bgu-tuebingen.de (A.D.); 2Ludwig Boltzmann Institute for Traumatology, The Research Center in Cooperation with AUVA, Donaueschingenstraße 13, 1200 Vienna, Austria; david.hercher@trauma.lbg.ac.at; 3Austrian Cluster for Tissue Regeneration, 1200 Vienna, Austria

**Keywords:** median nerve, grasping test, nerve repair, nerve crush, nerve transection, rat, animal, functional recovery, autograft, allograft

## Abstract

The rat median nerve model is a well-established and frequently used model for peripheral nerve injury and repair. The grasping test is the gold-standard to evaluate functional recovery in this model. However, no comprehensive review exists to summarize the course of functional recovery in regard to the lesion type. According to PRISMA-guidelines, research was performed, including the databases PubMed and Web of Science. Groups were: (1) crush injury, (2) transection with end-to-end or with (3) end-to-side coaptation and (4) isogenic or acellular allogenic grafting. Total and respective number, as well as rat strain, type of nerve defect, length of isogenic or acellular allogenic allografts, time at first signs of motor recovery (FSR) and maximal recovery grasping strength (MRGS), were evaluated. In total, 47 articles met the inclusion criteria. Group I showed earliest signs of motor recovery. Slow recovery was observable in group III and in graft length above 25 mm. Isografts recovered faster compared to other grafts. The onset and course of recovery is heavily dependent from the type of nerve injury. The grasping test should be used complementary in addition to other volitional and non-volitional tests. Repetitive examinations should be planned carefully to optimize assessment of valid and reliable data.

## 1. Introduction

Experimental models of peripheral nerve injury are essential to expand our understanding of neural regeneration following therapeutic interventions, which we continuously strive to improve, given the dramatic impact of peripheral nerve injuries on the affected patients’ quality of life [1,2,3]. The most frequently used rodent model to study peripheral nerve injury and regeneration is the sciatic nerve of the rat [4,5]. However, this model has some major disadvantages, such as the nerve’s mixed fiber type and its innervation of antagonistic muscles [6]. Most importantly, automutilation, e.g., the partial or total gnawing of the affected hind paw is frequently observed following sciatic nerve transection, assumed to be caused at least partially by neuropathic pain. This results in exclusion of animals with automutilations or inaccurate assessment of functional recovery which is often performed by means of walking track analysis or computerized gait analysis in such models of peripheral nerve injuries [6,7,8]. The rat median nerve model offers an interesting alternative [9]. As one of the main nerves of the brachial plexus, the median nerve provides muscular innervation to the flexor muscles of the paw and toes, e.g., the flexor digitorum superficialis and profundus muscles as well as sensory innervation for the medial aspect of the paw and digits I-III [10,11,12]. Rats with median nerve injury show a significant lower rate of automutilations, despite the onset of neuropathic pain following median nerve injury [13]. Additionally, translation of basic discoveries made in this preclinical model into clinical applications seems to be more appropriate, especially because upper extremity nerve injuries are more than three times more common than lower extremity nerve injuries [14,15]. Moreover, the brachial plexus of the rat is comparable in its components and branching to that of humans [16]. Another advantage of the rat median nerve model is the availability of several functional tests such as the staircase test developed by Montoya et al., the ladder rung walking test by Mets and Whishaw or computerized gait analysis, which was recently shown to be a valuable additional tool to assess functional recovery in rats with median nerve injury [7,11,17,18,19]. However, there is no single set of recovery evaluation standards recognized today [9]. The grasping test is a well-established method for functional assessment of median nerve injuries and was first introduced in 1995 by Bertelli and Mira [10]. In the original version of the assessment technique, rats were lifted by the tail and allowed to grasp the grid of a cage connected to an ordinary electronic scale. The animals were then gently lifted upwards by pulling at them at the base of their tails to induce the grasping reflex. Maximum grasping strength was measured in three consecutive trials and both maximal and median grasping strength were obtained. The use of the contralateral forepaw could either be prevented permanently for the course of the experiment via surgical denervation or by wrapping it with adhesive tape [10,20]. By using a bilateral median nerve injury model, stress due to fixation of the uninjured paw during the testing procedure could be avoided while the number of animals could be also decreased in accordance with the “3 R” of in vivo research. [10,20,21,22,23]. Papalia et al. modified the original approach in 2003 by using a small tower with triangle-formed bars and adhesive tape placed just under the grasping bars (Figure 1) [13]. This prevents the rats from introducing the entire paw under the bar and to use their wrist flexor muscles to lift the grasping bar. Moreover, with this setup, it is impossible for rats to walk on the triangle, a phenomenon which was frequently reported by Bertelli when he introduced the grasping test in 1995 [10,13]. In 2019, Hanwright and colleagues reported a new assessment technique named stimulated grip strength testing (sGST) [24]. For sGST the median nerve was stimulated percutaneously while the rat was in anesthesia, eliciting maximal tetanic contraction of the digital flexors, i.e., the superficial and profound flexor digitorum profundus muscles. The authors compared the results of this new evaluation method to volitional grip strength measurements as described above and isometric tetanic muscle force testing. They found that sGST demonstrated greater reliability and inter-trial repeatability than volitional grip strength measurement and showed a reliability comparable to isometric tetanic muscle force testing. Additionally, sGST allowed for the additional benefit of serial measurements.

In regard to the sciatic nerve model of the rat, a comprehensive systematic review published in 2021 has summarized the functional outcome following nerve injury and repair in this commonly used model of peripheral nerve injury [1]. Several authors have addressed the advantages and disadvantages of the rat sciatic nerve model [6,25,26]. A comprehensive review and meta-analysis of the course of functional recovery can serve as a valuable aid to researchers aiming to test a new therapeutic approach or reconstructive technique in this model when conceptualizing and planning their study. To the best of our knowledge, no study has yet comprehensively evaluated the course of functional recovery as assessed by the grasping test in different types of median nerve injury in rats. Therefore, this in this work, we aimed to report and summarize the course of functional recovery following crush injuries, transections and segmental resection of the rat median nerve by means of a systematic review and meta-analysis. Special attention was placed on the period until first signs of recovery (FSR) and maximal recovery of grasping strength (MRGS), respectively, became observable. Given that the grasping test is considered the gold standard method for functional examinations of median nerve injuries in rats, this work also discusses advantages and disadvantages of this assessment and its limitations. We conclude with an outlook on possible further applications of the method in the future.

## 2. Materials and Methods

The PRISMA (Preferred Reporting Items for Systematic Reviews and Meta-Analyses) guidelines for systematic review and meta-analysis were followed [27]. This review’s protocol was registered with the International Platform of Registered Systematic Review and Meta-Analysis Protocols (INPLASY) on the 15th of July 2022 and was last updated on 15th of July 2022 (Registration number: INPLASY 202270074). In accordance with the PRISMA guidelines, we aimed to summarize which models of rat median nerve injury have been evaluated with the grasping test so far. Moreover, we intended to compile which degree of functional recovery could be expected in every specific model. Therefore, we performed a comprehensive literature research by using the full-text archive PubMed and Web of Science. We used the search terms “rat”, “median nerve” and “grasping test” and their combinations, respectively. Figure 2 illustrates the flow chart of the systematic literature search according to PRISMA guidelines. Following title and abstract screening, the full texts of possibly eligible studies were retrieved and reviewed in regard to the predefined inclusion and exclusion criteria (Figure 2). Of note, only such studies were included which featured at least two postoperative time points at in which functional recovery was evaluated. Excluding criteria involved the use of models of partial nerve injuries, spinal cord and spinal root injuries. After application of these criteria, 47 papers utilizing the grasping test to evaluate functional recovery in rats with median nerve injuries were included in this review. These studies’ publication dates ranged from 1995 to 2021.

The included studies (Table A1) were then clustered in four main groups based on the respective nerve injury model used and reconstructive approach examined by the investigators: (I) crush injuries (axonotmesis), (II) nerve transection (neurotmesis) and subsequent end-to-end repair with or without adjuvant therapies, (III) nerve transection (neurotmesis) and subsequent end-to-side neurorrhaphy with or without adjuvant therapies, and (IV) segmental nerve injuries (neurotmesis) reconstructed with Isogenic, acellular allogenic or other types of grafts with or without adjuvant therapies.

For the overall number of included studies in general and each study in particular, the following data were extracted: (1) name(s) of the rat strain(s) used, (2) included rats per study, (3) type of nerve defect and (4) length of the isogenic or acellular allografts. We also calculated the time at first signs of motor recovery (FSR, in days) based on the data extracted from each study. We defined the FSR as the time interval that has passed between the experimental nerve injury and the postoperative time point which provided the first measurable grasping strength or visually detectable sign of reinnervation of the extrinsic flexor muscles, e.g., voluntary toe flexion. In case the respective data were reported and could be extracted from the manuscript, we also calculated the maximal recovery grasping strength (MRGS, in days), defined as the period between the experimental nerve injury and the postoperative time point at which maximum values of grasping strength during respective postoperative period, could be observed.

### Data Analysis

Data are represented in absolute values as boxplots with the mean separately indicated. No further statistical comparisons between groups were performed due to the heterogeneity of the individual studies in regard to data reporting, length of the postoperative follow-up and the number of follow-up examinations.

## 3. Results

A total of 1758 animals, mostly female *Wistar* rats, were included in all studies which were considered eligible for this work. The respective rats’ strain and sex are summarized in Table 1. Automutilations, joint contractures or weight loss due to insufficient food intake were not reported in any study. In total, 46 rats (2.6% of the studied population) died during the respective observation period. The only reported reason was hypovolemic shock due to hematoma formation [28].

The most frequently used nerve injury model (Table 2) among all included studies was segmental neurotmesis, repaired by means of an isogenic nerve graft and compared to allograft +/− adjuvant therapies (44.7%). Non-segmental neurotmesis, e.g., nerve transection repaired with end-to-end neurorrhaphy +/− adjuvant therapy, was the second most frequently used model (21.2%).

The earliest manifestations of first signs of motor recovery (FSR) (Figure 3) were observable in the crush injury group (n = 165) starting from the eighth postoperative day (median = 10, mean = 10.4) with constant motor performance from postoperative day 21 to 32. Rats in the other groups (II-IV) showed an inhomogeneous onset of functional recovery. The mean time point of FSR in the End-to-End repair group was 23.3 days in group II (median = 21), 75.4 days in group III (median = 70) and 47 days in group IV (median = 47, mean = 42).

In case of experimental nerve grafting (Figure 4), grafts with lengths of under 15 mm were most frequently used (52.4%, 11/21 articles). The FSR ranged between 12 and 60 days with an average of 41 days (median = 42). Nerve reconstruction using grafts with lengths under 25 mm resulted in a mean FSR of 45.3 days (median = 37). Grafts with lengths of more than 25 mm were used in five studies, (22% of all studies). Out of these articles, four studies reported a FSR of 38 to 150 days postoperatively with a mean of 68 days (median = 42). In one article, FSR was not reported.

In studies which used autografts for nerve reconstruction (Figure 5), the earliest FSR was reported on day 28. The mean FSR following nerve repair by autografts was 52.6 days (median = 42). In case acellular allogenic grafts or other grafts were used, the FSR ranged between postoperative day 30 to 90 (mean = 69.8, median = 77).

In groups in which median nerve transection was performed (group II and III) the MRGS (Figure 6) could be extracted from 16 out of 21 articles (76.2%). Rats undergoing end-to-end repair showed a mean MRGS of 99.9 days (median 91 days), whereas in end-to-side-repair groups a mean MRGS of 137.2 days (median 112 days) was observable (Figure 6).

## 4. Discussion

### 4.1. Use of the Grasping Test to Assess Functional Recovery in the Rat Median Nerve Model

The advantages of the grasping test as assessment method for functional recovery following murine median nerve injuries have been pointed out previously by other authors [10,13,22]. Considering that functional recovery is the most important criterion to determine the effect of any neurogenerative treatment or reconstructive technique [29], the grasping test is a very specific functional assessment for median nerve injuries, because finger flexion is predominantly mediated by the median nerve in rats [20]. The grasping test indicates the precise day on which functional recovery begins without the need of complex preparation or anesthesia [10,22]. Moreover, the grasping test is widely used to evaluate the motor function of median nerve innervated muscles and, therefore, a plethora of comparative data is available [11].

Alternative methods, such as sGST or noninvasive electrophysiological recordings, require anesthetized animals, so volitional movement cannot be measured in this context [22,24]. On the other hand, non-volitional testing is more precise with fewer fluctuations in functional improvement after nerve repair, and also has the advantage of detection of earlier signs of nerve regeneration compared to the conventional grasping test. As pointed out previously, this technique also allows for serial measurement without the imminent decrease of animal motivation to participate in the procedure [24]. Furthermore, complementary methods, such as electrophysiological assessments, can evaluate subtle differences during the process of nerve regeneration, although differences in functional recovery, e.g., grasping strength, might be missing [11,30,31,32]. However, the need of repetitive anesthesia is still problematic in regard to animal welfare [33,34].

Besides its benefits, the grasping test has some major limitations. First, its results heavily depend on the speed with which the animal is moved towards the ceiling to provoke the grasping reflex. No method has been developed so far to standardize this movements; therefore, it is mandatory that the measurements are performed by the same investigator to reduce interexperimenter variability [7,13,20]. Second, certain rat strains such as rats of the Lewis strain display a lack of motivation in participating in the grasping test when it is per-formed repeatedly, leading to fluctuating performances or even unreproducible results [7,22]. Next, the grasping test was shown to address a sensory-motoric reflex circuit, so that the inference of altered sensation of the front paws, the experience of neuropathic pain or other sensory disturbances might influence its results [7]. Last but not least, the assumption that the flexor digitorum superficialis and profundus muscles are predominantly innervated by the median nerve with no or only insignificant contribution of the ulnar nerve is still a matter of scientific debate [7]. Therefore, we deem it reasonable to further evaluate the specific innervation pattern in detail, e.g., by means of retrograde labelling or other techniques which allow to unveil the contribution of both the median and ulnar nerve to innervation of the long finger flexor muscles. In this context, interindividual and inter-strain variations should also be considered.

### 4.2. Functional Recovery in the Context of the Respective Nerve Injury Model Used

Axonotmesis [35], e.g., discontinuity of the axon which can be induced by a crush injury of a nerve, is a widely used experimental approach in murine models, especially the sciatic nerve model of the rat [20,36,37]. In our analysis, the “median nerve crush injury”-group showed a FSR which ranged between postoperative day 8 and 12. Ronchi et al. presented a standardized median nerve model for axonotmetic lesions with a degree of functional recovery of 75% as compared to preoperative grasping strength values at day 28 [36]. Other authors reported a MRGS of three or four weeks following median nerve crush injury [19,38]. It should be mentioned that crush injuries show fast recovery with very good results in general and the surgical interventions are almost never necessary [4,7,35,36]. Therefore, the FSR and MRGS values our study reports for this specific subgroup of median nerve injury models are not particularly surprising. Besides its fast recovery ad integrum, the crush injury models have other substantial advantages. A standardized crush injury can be introduced with basic laboratory equipment and no microsurgical skills are needed. However, it must be noted that the fast speed of nerve regeneration following axonotmesis also has disadvantageous implications. Due to the fast speed of nerve regeneration, and, therefore, functional recovery in rodents, differences in nerve regeneration which might relate to the respective experimental treatment approaches or therapies are difficult to detect, hampering translational research in this small animal models [26,39].

Our work shows that in case of neurotmesis, e.g., transection of the entire nerve including its enveloping sheaths, of the rat median nerve, functional recovery ad integrum is highly unlikely to occur. Best functional outcomes were observable following direct nerve coaptation in end-to-end technique, mirroring results of primary repair of the median nerve in humans [40]. Like sciatic nerve models, second best outcomes were obtained following autograft repair of segmental nerve injuries [1].

Nerve repair by means of end-to-side coaptation resulted in a prolonged recovery time which was almost three times slower longer than following end-to-end coaptation (Figure 6). In our opinion, the selection of donor nerves should be considered as important factor in case of end-to-side neurorrhaphy, i.e., nerve transfer, as was shown by other authors [41]. Papalia et al. noted that also antagonistic nerves, e.g., the radial nerve as donor for the median nerve, can be used as donors for end-to-side neurorrhaphy, but the duration of the postoperative observation period must be increased [42]. The surgical paradigm of nerve transfers has gained increasing popularity during the last two decades [43] with new concepts such as the “babysitter” [44,45] end-to-side nerve transfer or the “supercharge” [46] reverse end-to-side nerve transfer. Now, as ever, clinical and preclinical research on nerve transfers is a main area of interest within the scientific community of peripheral nerve researchers [47,48]. We, therefore, advise that preclinical researcher interested in this area should carefully consider the time period until first signs of functional recovery can be expected when the rat median nerve is used as an experimental model. The use of autologous nerve grafts is considered the gold-standard method for reconstruction of segmental nerve defects [49,50,51]. This is also mirrored by the results of our analysis as the functional outcome of autologous nerve grafting was superior compared to the results following nerve repair by means of other grafts (Figure 4 and Figure 5). Studies with graft length greater than 24 mm were not compared because just one of them investigated grafts other than autografts and, in this investigation, rats failed to show functional recovery [49]. Defects larger than 25 mm are difficult to create surgically due to the small forelimb length of rats [52]. Sinis et al. described a model for nerve regeneration across a 40 mm gap in the rat by using a cross-chest median nerve transfer [53]. Bertelli et al. accommodated the graft by tunneling it over the dorsal side of the triceps muscle [54]. These studies had in common that rats treated with such long grafts were able to recover motor function, e.g., grasping, in general, but postoperative follow-up times of up to 12 months are necessary to adequately monitor the course of functional recovery following nerve grafting over of defects which such extensive size. Therefore, in principle, larger defects can also be examined by means of the grasping test, but this will require longer postoperative follow-up and vice-versa higher expenses in regard to animal housing, etc. [23,53,54,55,56]. However, the influence of the graft length on functional recovery remains a subject of scientific debate, especially when comparing preclinical rodent models to human patients [52,56]. Variation in lengths of nerve autografts and allografts were shown to significantly influence the course of functional recovery [57,58,59]. In our analysis, shorter graft with lengths below 15 mm were not associated with faster recovery compared to graft with lengths below 25 mm (Figure 4). The heterogeneity among the studies included in our analysis (e.g., different follow-up intervals, different treatments, etc.) could, in our opinion, serve as a valid explanation for the observed irregularities and differing results. Moreover, the level of the nerve damage (upper part or middle part of the brachium) and, therefore, the distance the regenerating nerve has to overcome, as well as the defect length, influences the functional recovery [60,61]. In any case, a tension free-coaptation is paramount to achieve optimal functional results, given the detrimental effects of tension on intraneural perfusion [62,63,64,65,66].

The respective injury model is of high importance regarding the planning of a preclinical study of peripheral nerve injury and repair. The same applies to the amount and frequency of postoperative evaluations of functional recovery following experimental nerve surgery. With this work, we provide researchers in the field of preclinical peripheral nerve research a valuable guide in the style of a reference book which will help to choose the appropriate framework conditions for their respective study when utilizing the median nerve model of the rat.

### 4.3. Limitations of This Work

Concerning the studies included in our review, there was no standardized setup of postoperative evaluation time-points by means of the grasping test. Given the heterogeneity of observation periods, as well as time points of functional assessments and follow-up examinations, only a limited analysis could be performed. The overall length of postoperative follow-up in the included studies differed from 12 to 48 weeks; therefore, there was also a limitation in regard to the calculation of the MRGS and its comparability between the different included studies. In summary, functional nerve regeneration could be observed in all the nerve injury types mentioned before, striking differences to the spatial course of functional recovery were also apparent. A very important tool for the interpretation of data in motor function is the stagnation of recovery after nerve injury. Referring to the rat median nerve model, our work indicates that stagnation of recovery can be expected between the 13th and 18th postoperative week, regardless which injury model is used [67,68].

## 5. Conclusions

Our work revealed that the earliest signs of functional recovery as assessed by the grasping test in rats can be expected in models of median nerve crush injuries between 8 and 12 days. Primary end-to-end repair results in better functional outcomes than end-to-side repair with a median recovery time of 21 days compared to 70 days. For reconstruction of segmental nerve defects, autologous grafts are superior to other grafts. Based on our results, the critical length for functional recovery in the rat forelimb is a nerve graft with a length of about 20–24 mm. In such cases, first signs of functional recovery might be observable after more than 3-4 months, by the earliest. The main benefits of the grasping test, e.g., specific assessment of the median nerve’s target organs by means of a simple method have been outlined by us and other authors. However, depending on the specific research question, other complementary evaluation methods for functional recovery, computerized gait analysis should be considered for an experimental study utilizing the rat median nerve model. Follow-up examinations should be limited to the minimum amount required so that the animals do not lose motivation to participate in investigations, especially when rats of the Lewis strain are used. For neurotmesis models, the over-all length of the observational period should exceed 100 days in case that maximal functional recovery should be observed. Overall, this review is intended for experimental researchers interested in the field of peripheral nerve injuries and nerve regeneration. It provides valuable information for planning and conducting experiments involving the rat median nerve model. Special emphasis is placed on the grasping test, its advantages and its limitations. By applying the above instructions, sources of error can be avoided in order to achieve reliable and valid results.

## Figures and Tables

**Figure 1 biomedicines-10-01878-f001:**
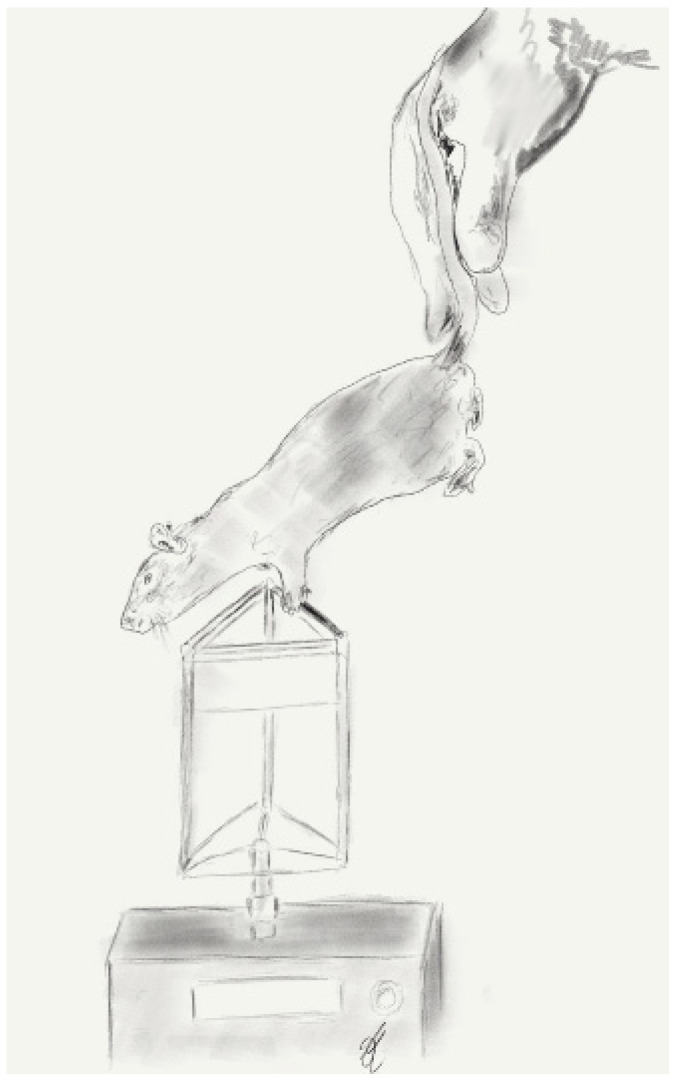
Schematic representation of the grasping test modified by Papalia et al., 2003 [13].

**Figure 2 biomedicines-10-01878-f002:**
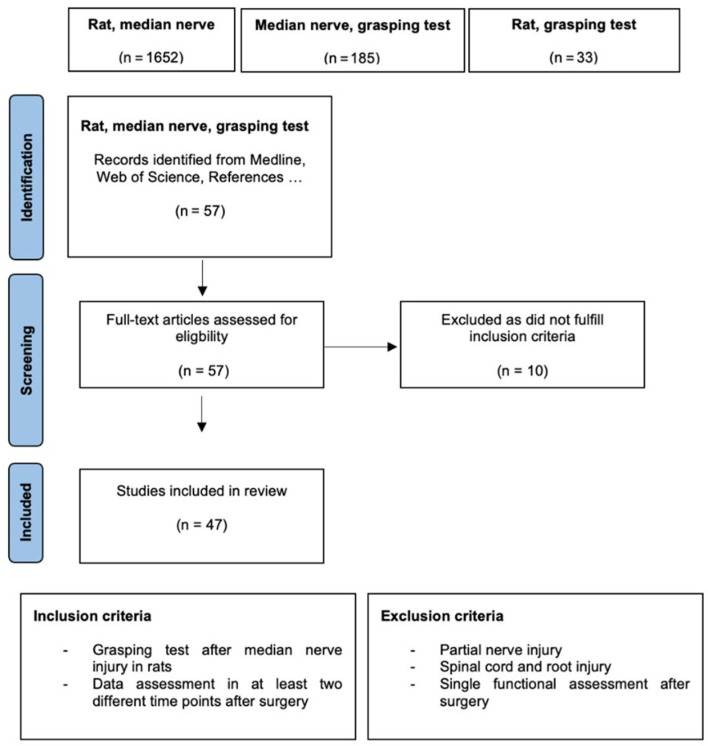
Flow chart of the systematic literature search according to the PRISMA guidelines [27]. The flow chart depicts the selection process of the retrieved studies in chronological order. We used the search terms: “rat”, “median nerve”, “grasping test” and their respective combinations.

**Figure 3 biomedicines-10-01878-f003:**
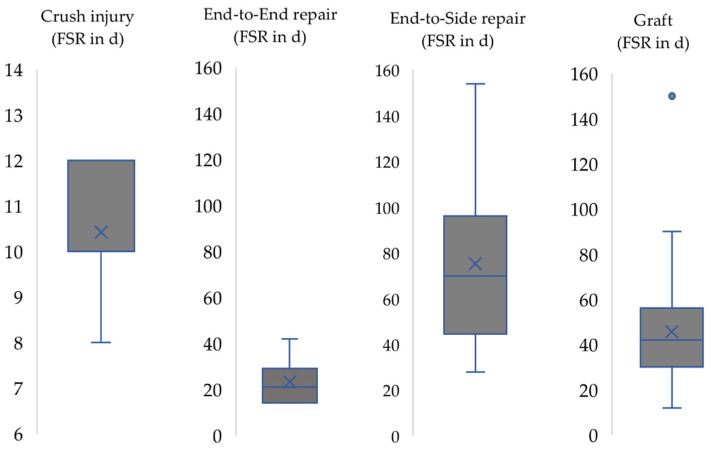
First signs of motor recovery (FSR) in group I-IV. X: mean. Horizontal line: median. dot: outlier.

**Figure 4 biomedicines-10-01878-f004:**
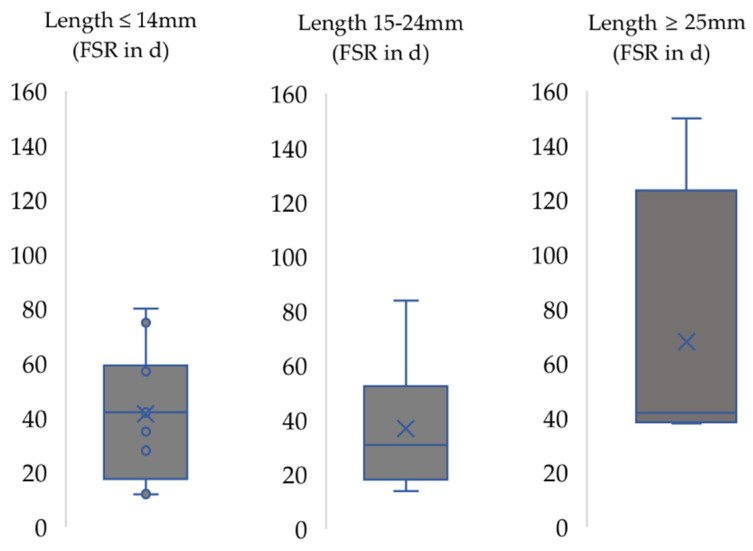
First signs of motor recovery (FSR) following nerve repair with graft length ≤ 14 mm, 15–24 mm and ≥ 25 mm. x: median. horizontal line: mean.

**Figure 5 biomedicines-10-01878-f005:**
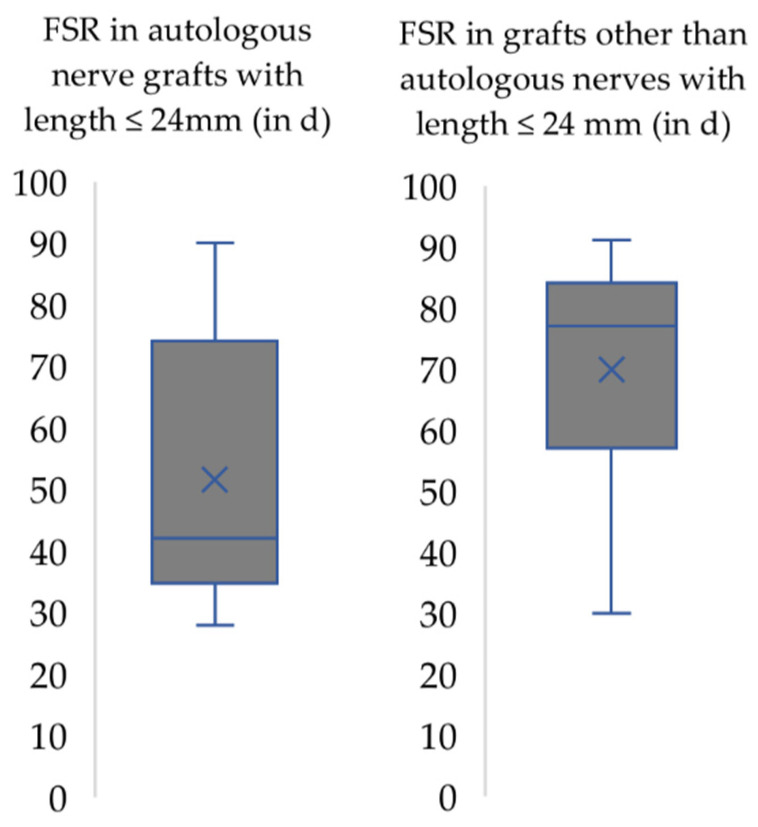
First signs of motor recovery (FSR) following repair of the median nerve with autologous nerve grafts (**left**) compared to other grafts (**right**) with length ≤ 24 mm. x: median. horizontal line: mean.

**Figure 6 biomedicines-10-01878-f006:**
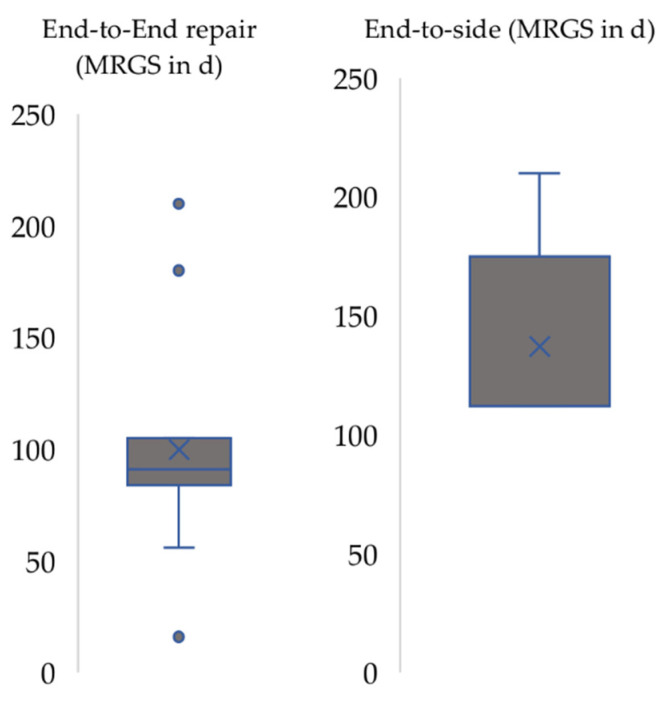
Maximal recovery grasping strength (MRGS) in end-to-end repair and end-to-side repair. x: median. horizontal line: mean.

**Table 1 biomedicines-10-01878-t001:** Strains and sexes of all rats included in the 47 articles which were found to be eligible for this review. ***** Note that in one included study two different of rat strains were used [19].

Strain and Sexes *	Female	Male
*Wistar*	23	4
*Lewis*	5	3
*Lister Hooded*	1	0
*Sprague Dawley*	12	0

**Table 2 biomedicines-10-01878-t002:** Groups and nerve models’ distribution in 47 articles (two of them used different models, so this resulted in 49 models in total). I: Crush injury II: Transected nerve with End-to-End repair and/or adjuvant therapies III: Transected nerve with End-to-Side repair and/or adjuvant therapies IV: Isogenic, acellular allogenic or other grafts.

Group	
I	7 (14.9%)
II	13 (21.2%)
III	8 (17%)
IV	21 (44.7%)

## Data Availability

The datasets analyzed during his systematic review are available from the corresponding author in reasonable request.

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
