# Peer review of "The Grasping Test Revisited: A Systematic Review of Functional Recovery in Rat Models of Median Nerve Injury"

_biomedicines, 2022, doi:10.3390/biomedicines10081878_

Round 1

Reviewer 1 Report

The authors used the PRISMA model to perform a systematic review of functional recovery in various rat models of median nerve injury. Forty-seven studies were included in the review evaluating four primary models of nerve injury. The authors calculated first signs of motor recovery, which was then broken down to further assess graft length and type. They also assessed maximal recovery of grasping strenght via end-to-end repair or end-to-side repair. Overall, the study is well written and provides quality information to help with study design for researchers working with rat models of median nerve injury. 

Minor points: 

1. The authors introduced abbreviations such as first sign of motor recovery (FSR), which wasn't consistenly carried throughout the manuscript.

2. In the conclusion, the authors address what was learned from the study  and how they think this knowledge should best be applied to the field (Lines 337-353). They should also address who their desired audience is and how this new knowledge advances the field.

Author Response

Dear Reviewer No.1,

Thank you very much for your helpful comments and suggestions.

1. Please excuse this mistake of ours. We went through the manuscript again and corrected these inconsistencies according to your remark.
2. Thank you for your suggestion. We added a pass on line 360 to 365 to address our audience and described in which manner our study could be helpful. 

Reviewer 2 Report

The Review conducted by Lauer and colleagues systematically expounded on recovery in rat models of median nerve injury. Different models are compared, here is a nice bioinformatics review. The author addressed this review as comprehensive and systematic references, most of the cited papers are within 5 years, so this work is very convenient for readers to read and can help readers better understand related fields. This is a qualified review.

Author Response

Dear Reviewer No. 2,

Thank you very much for your comment. We are pleased to hear that our study is a qualified review and really appreciate your feedback.

Additional note to all reviewers:
Please note that a registrations statement and registration number was added (lines 106-110).